# Enhancing Organizational Resilience through Mindful Organizing

Siriwut Buranapin ⬤, Wiphawan Limphaibool *⬤, Nittaya Jariangprasert and Kemakorn Chaiprasit

Department of Management and Entrepreneurship, Faculty of Business Administration, Chiang Mai University, Chiang Mai 50200, Thailand
* Correspondence: wiphawan_l@cmu.ac.th; Tel.: +668-0001-2211

**Abstract:** Organizational resilience and mindfulness are inextricably connected and have specific characteristics related to responding to challenging events. This mixed-method research study aimed to explore the relationship between mindful organizing and organizational resilience. A qualitative critical incident analysis was conducted with executives to explore insights into mindfulness and resilience at the organizational level. Using the analysis of a moment structures (AMOS) program, the structural equation modeling method was employed to assess the relationships between mindfulness, mindful organizing, and organizational resilience. A total of 639 usable cross-sectional questionnaires from diverse organizations in Thailand were used for data analysis. The findings of the current study reveal that mindful organizing has a positive influence on organizational resilience. This paper discusses the implications and limitations of these findings, along with suggestions for future research.

**Keywords:** organizational resilience; mindful organizing; organizational mindfulness; mindfulness

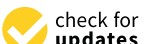



## 1. Introduction

Organizations inevitably confront unexpected events under greater pressure than they have experienced in the past. Rapidly evolving external and internal environmental factors impact organizations' survival and continuality: for example, the COVID-19 pandemic has directly and significantly impacted a large number of organizations. During the pandemic, many businesses closed temporarily or permanently, with the resulting job losses significantly impacting the economy. Inadequate preparation and a flawed recovery plan are reasons many organizations face crises and go out of business [1].

Organizations must, therefore, respond quickly and develop the capacity to be resilient, which enables them to be prepared for unexpected events and recuperate from crises. Organizational resilience refers to organizations' ability to respond, recover, and develop beyond their competitors [2]. For business sustainability, resilient organizations can respond, recover, and grow in the case of disruptions. Kantabutra suggests that developing a perseverance culture and practicing resilience within an organization enhances corporate sustainability [3]. Moreover, business sustainability contributes to organizational resilience [4]. Organizational resilience and corporate sustainability are critical capabilities for business continuity management [5]. However, organizational resilience remains a 'black box', where both the input and output are observable, but the process between them is unknown [6]. It is vital to identify the attributes and activities which support an organization in enhancing its resilience in a world where change happens rapidly.

Mindfulness, an attribute of consciousness, has attracted the attention of many businesses and has become an essential factor in environmental change and associated sustainability research [7]. Hyland et al. suggest that mindfulness may help employees to cope with organizational change as it relates to workplace achievement and success. Mindfulness positively impacts workplace outcomes, including resilience; creativity; productivity; work

engagement; and reduced conflict, absenteeism, and turnover [8]. Many business scholars have demonstrated the importance of mindfulness in recovery, and both resilience and mindfulness have similar characteristics in responding to challenging occurrences. Mindful organizing is the collective capacity of members in an organization to attend to and act on errors and unexpected circumstances [9].

However, fewer studies have emphasized the association between mindfulness and organizational resilience [10]. The integration of mindful organizing and organizational resilience posits a better understanding of how organizations recover and develop following unanticipated events. Therefore, this study aims to explore the correlation between mindfulness and organizational resilience and whether the impact of mindfulness at the individual and/or organizational level can enhance organizational resilience. It contributes to the theories of organizational resilience and high-reliability organizations (HROs), which relate to mindful organizing. The results indicate how organizations develop their ability to maintain function and structure in the face of adversity through the core elements of mindfulness and the critical factors of organizational resilience.

In the next section, we define mindfulness, mindful organizing, and organizational resilience. We further review the literature on how individual mindfulness promotes mindful organizing and explore the possibility that greater mindful organizing is beneficial to organizational resilience. Based on this review, we note the potential for enhancing organizational resilience through mindfulness. Following this, we explain the theoretical framework, conceptual model, and methodology hypothesis and discuss the results and the need for additional research on these topics.

## 2. Literature Review

### 2.1. Mindfulness

Recently, mindfulness research has received considerable attention and growth in many fields from both a theoretical and practical perspective. Sutcliffe et al. identify two focuses of mindfulness: individual and organizational [11]. The term 'mindfulness' typically refers to individual mindfulness. In the Eastern concept, mindfulness originates from the Pali word sati, which indicates the presence of the mind and refers to awareness, attention, and remembering in the form of consciousness [12]. Mindfulness is considered to be conscious awareness through nonjudgmentally paying attention in the present moment [8,13,14].

There is another Western perspective of mindfulness. Langer argues that mindfulness is different from mindless behavior; mindfulness is a state of being present and wakeful, which leads to greater sensitivity to the environment, more openness to new information, the creation of new categories for perception, and enhanced awareness of perspective in problem-solving [15,16]. Based on Langer's Western concept of mindfulness, the idea of high-reliability organizing is developed, which leads to the concept of mindful organizing as infrastructure in the theory of HROs in this study [17].

### 2.2. Theory of High-Reliability Organizations

The concept of HROs originated from a team of researchers at the University of California, Berkeley, in 1984, who argued that error-free performance is created by an active search for reliability [18]. They noted that three organizations (i.e., the US Air Traffic Control system; Diablo Canyon, an electric company that operates a nuclear power station and an electricity distribution system; and the US Navy nuclear aircraft carrier operations) operate effectively in complex technical environments where their errors could have destructive effects. HROs are organizations that operate in complex, highly risky areas and potentially must cope with catastrophic consequences when failures occur. Weick and Roberts suggest that HROs have accomplished collective mental processes, including information processes, heedful action, and mindful attention. They propose the concept of a collective mind and argue that increased attentiveness and mindful comprehension decrease the likelihood of errors in an organization [19]. Weick et al. created a mindfulness infrastructure at the level

of collective or organizational mindfulness. The theory of HROs points to mindfulness as the infrastructure of HROs. High-reliability organizations attempt to be error-free under complex and high-risk systems. They argue that the success of HROs lies in handling unexpected events by acting mindfully; that is, organizations are better able to notice unpredictable events, develop, focus on containing and resilience, and quickly recover from system functioning [9,20,21]. Figure 1 presents the five key characteristics of HROs.

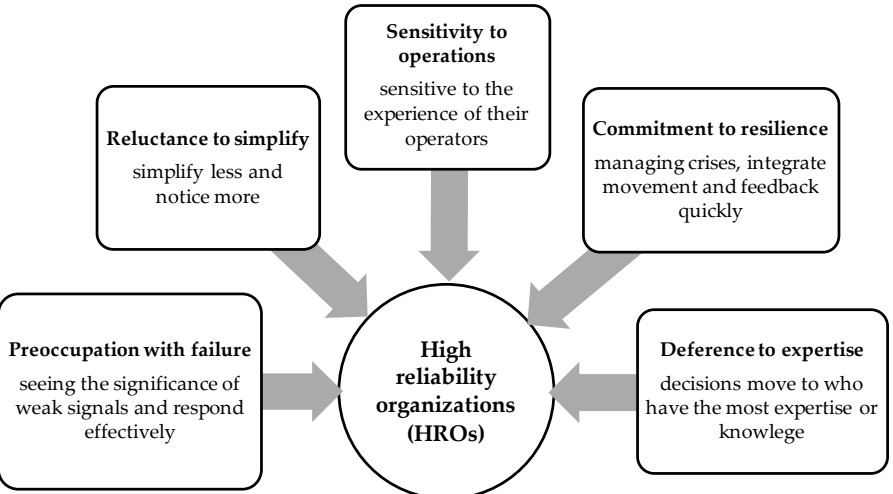

**Figure 1.** Five characteristics of high-reliability organizations, adapted from Weick and Sutcliffe, 2001 [21].

The theory of HROs is connected to mindfulness and resilience. This study collects empirical evidence on mindfulness, as it is a key element of organizational resilience in both high-reliability and general organizations, which has yet to be examined.

### 2.3. Mindful Organizing

Mindful organizing is the collective capacity of members to pay attention to context and act on errors and unexpected circumstances. Attention to context and acting on errors is generated through the five key characteristics of HROs (i.e., preoccupation with failure, reluctance to simplify, sensitivity to operations, commitment to resilience, and deference to expertise) [9]. In the theory of HROs, mindful organizing is the infrastructure that supports HROs' exceptional performance [22]. Mindful organizing is not an internal process in the minds of individuals but rather a set of social and organizational processes that focus on organization members' dependence on continuous real-time communications and interactions, leading to corrective action [23]. The five characteristics of HROs are conceptualized as predictors of mindful organizing. Additionally, mindful organizing is significantly related to environmental and resource sustainability [24].

### 2.4. Organizational Resilience

Resilience is typically acknowledged as the ability to recover and thrive in the face of adversity. At the organizational level, organizational studies have discussed and increasingly focused on resilience as a concept of survival, which requires organizations to be adaptable and flexible to respond to uncertain environments [25]. Resilience refers to the characteristics of organizations that handle challenging situations by responding and recovering rapidly or developing more than others [2]. Organizational resilience is an organization's ability to recover from the impact of change. It is viewed as an organization's qualities that enable it to cope with, adapt to, and recover from an unexpected event [9,26,27].

Factors and elements that enhance organizational resilience have been presented in previous studies in many ways, and the factors that foster organizational resilience are not apparent. Most studies have employed a case study or grounded theory rather than

an empirical study. McManus conducted a qualitative study and semi-structured interviews to explore organizational resilience and suggested three factors of organizational resilience: situational awareness, the management of keystone vulnerabilities, and adaptive capacity [28,29]. This study applies the factors in the work of McManus as fundamental to organizational resilience. McManus's concept merges the cognitive processes with mindfulness and the organizational capacity to adapt, which is consistent with the theoretical framework of this study.

## 3. Theoretical Framework, Research Model, and Hypotheses

### 3.1. Theoretical Framework

This study employs Langer's concept to define mindfulness (i.e., novelty seeking, novelty producing, flexibility, and engagement). The theoretical foundation for this study is HROs and mindful organizing (i.e., preoccupation with failure, reluctance to simplify, sensitivity to operations, commitment to resilience, and deference to expertise), which comprise the core infrastructure of HROs [9]. Mindful activity is a dynamic process that dictates behavior in group settings. As applied to this study, mindful organizing is expected to predict organization-level resilience. The relative overall resilience (ROR) model employed by this study comprises three factors: (1) situational awareness, (2) the management of keystone vulnerabilities, and (3) adaptive capacity [28]. Figure 2 presents the theoretical framework that demonstrates the theories relevant to this study.

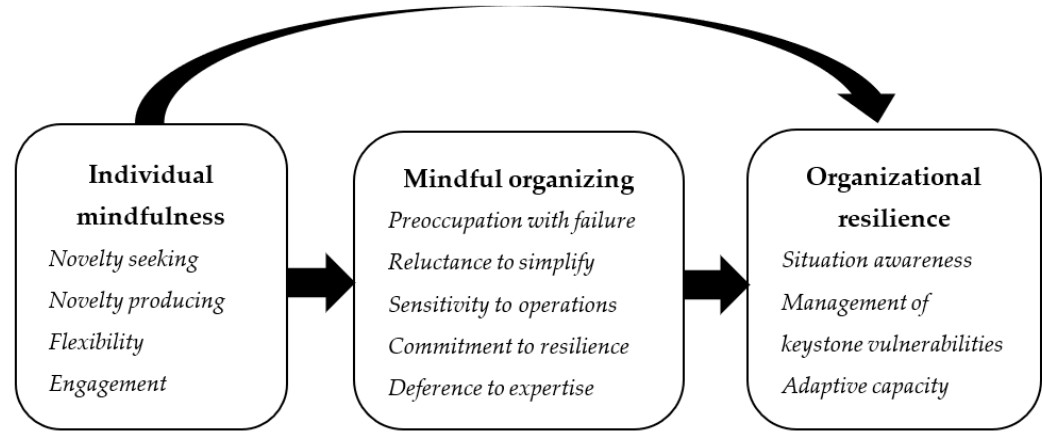

**Figure 2.** Theoretical framework.

### 3.2. Research Model and Hypotheses

This study investigated the organizational resilience capability, which relates to mindfulness at the individual and organizational levels. We posited that the individual mindfulness of members leads to the collective capability of members (i.e., mindful organizing), which enhances organizational resilience. The relevant theories and the theoretical framework (see Figure 2) led to the research hypotheses in this study.

Mindful organizing was developed in individual mindfulness as a foundation. It is a social process that becomes a collective capability through interactions among individuals [15,22]. Individual and mindful organizing share an emphasis on increased attention to the present moment situation and acting on what they notice [9,23,30]. This study attempts to confirm the link between individual mindfulness and mindful organizing theoretically. The first hypothesis is as follows:

**Hypothesis 1 (H1).** *Individual mindfulness has a positive effect on mindful organizing.*

Mindful organizing is the quality of attention at the level of the collective organization, which relates to what people decide to do with what they notice when facing unexpected events [9]. At the organizational level, five characteristics of HROs were identified as the

necessary elements of organizational resilience [31,32]. Oliver et al. define organizational mindfulness as a quality of an organization that reliably and effectively operates in the face of challenging conditions, and they find a significant positive correlation between mindfulness, resilience, and performance [33]. This study explored the relationship between mindful organizing and organizational resilience to contribute to the theory of HROs. The next hypothesis is as follows:

**Hypothesis 2 (H2).** *Mindful organizing has a positive effect on organizational resilience.*

There is no empirical evidence of the relationship between individual mindfulness and organizational resilience. However, certain arguments exist regarding the positive impact of mindfulness on organizational change, sustainability, outcomes, performance, leaders' decision-making, and success [7,22–24]. The third hypothesis is as follows:

**Hypothesis 3 (H3).** *Individual mindfulness has a positive effect on organizational resilience.*

Most studies on mindfulness at the collective organizational level have been qualitative [11]. Hence, no evidence indicates mindful organizing as a mediator of constructs. There is a need for more research on mindful organizing at work, especially using quantitative methods. This study is the first attempt to explore the relationship between individual mindfulness and organizational resilience by using mindful organizing as a mediator. The fourth hypothesis is as follows:

**Hypothesis 4 (H4).** *Mindful organizing significantly mediates the relationship between individual mindfulness and organizational resilience.*

Figure 3 shows a path diagram for the causal relationships between the three constructs in enhancing organizational resilience: individual mindfulness ($\xi 1$), mindful organizing ($\eta 1$), and organizational resilience ($\eta 2$). Mindful organizing and organizational resilience are endogenous latent variables, while individual mindfulness is an exogenous latent variable. The SEM for this mediation model is given by:

$$\eta 1 = \beta_{\xi 1 \eta 1} \xi 1 + \zeta_{\eta 1}$$

$$\eta 2 = \beta_{\eta 1 \eta 2} \eta 1 + \beta_{\xi 1 \eta 2} \xi 1 + \zeta_{\eta 2}$$

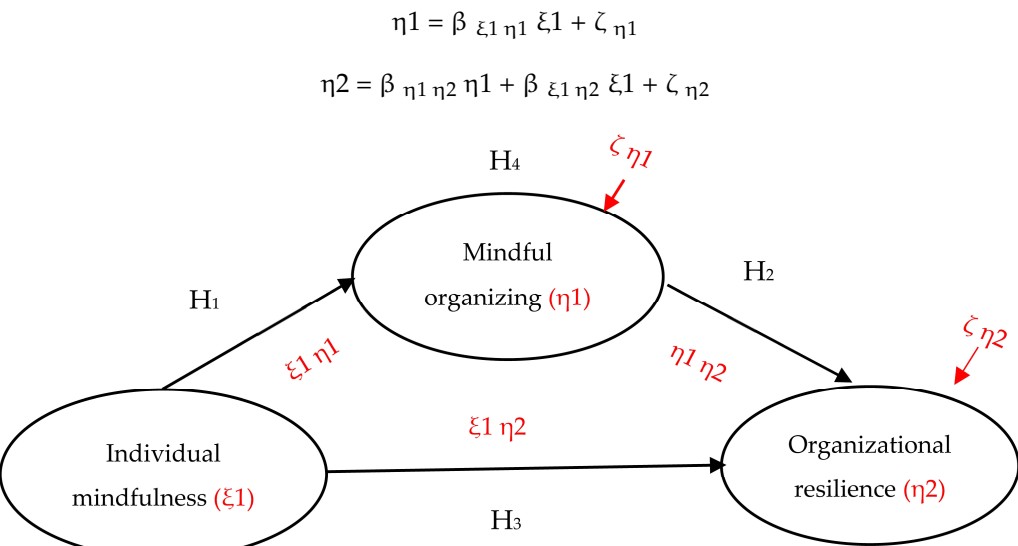

**Figure 3.** Research model and hypotheses.

## 4. Research Methodology

This study used an exploratory mixed method which involved two data collection and analysis phases. First, qualitative data collection and analysis were employed using the critical incident technique (CIT) to explore the correlations between constructs, individual

mindfulness, mindful organizing, and organizational resilience. Second, the quantitative data collection and analysis explored the measurement and analysis of causal relationships between these constructs.

*4.1. Qualitative Study*

In the first phase, the CIT was selected to confirm the theoretical framework of this study. The CIT is a widely used qualitative method and an effective exploratory and investigative tool. It is a set of procedures for gathering and analyzing reports of incidents and important facts concerning behavior in a defined situation, as well as a well-established qualitative research methodology for exploring significant experiences to understand problem-solving behavior better [34].

4.1.1. Participants and Procedures

The participants were purposively selected via the criterion-based selection method and interviewed using the CIT. The 10 participants included high-level executives who worked for organizations based in Thailand, including hospitals, the investigation division of the Royal Thai Police, and the liquid petroleum gas terminal, while general organizations included manufacturing, wholesale, retailing, service, media, and educational institutions. Data were collected via in-depth and semi-structured interviews, which were approximately two hours in length. Each participant was given a definition of the critical organizational incident. The participants were asked to recall the most critical incident of the organization that they could clearly remember and to share the cause, procedure, and summary of the event. If the essential factors of individual mindfulness and mindful organizing, which enhance organizational resilience, were not mentioned in the interviews, additional questions sought to access the constructs in this study, such as 'How do you adapt to novel changes during an unexpected event?'; 'Did you or your staff notice unusual signals before the event? How?'.

4.1.2. Data Analysis

All interviews and discussions using the CIT were recorded and transcribed to be interpreted for meaning. Transcriptions were completed word-for-word using audio or video recordings and included non-verbal expressions. Data were analyzed using thematic analysis, a flexible method for identifying, analyzing, and reporting themes, providing a rich and detailed description of these data [35]. This study employed a deductive approach to thematic analysis based on the theory of mindfulness, mindful organizing, and organizational resilience. The deductive approach allowed for a focus on specific aspects of these data to achieve the study objectives. After coding, we found associations between constructs in the interviews as follows:

'We faced a severe crisis. If we could not sell to this big customer, our company would have closed. I tried to find the solution, negotiate and compromise with the customer.' (This represents individual mindfulness—novelty seeking.) 'If we don't solve a problem, ignoring it and waiting hopefully will cause significant damage.' (This represents mindful organizing—preoccupation with failure.) The managing director of the manufacturing company described what he did when faced with a significant crisis: 'We learn the strong and weak points of staff and train them as appropriate.' (This represents organizational resilience—situational awareness.) He also presented the company's situational awareness of roles and responsibilities. He stated that he had knowledge of the roles and responsibilities of the staff in his organization.

'When we work, police must use judgment, knowledge, and capability, and they should notice things that are not normal.' (This represents individual mindfulness—engagement.) The deputy commissioner, an executive of the Royal Thai Police, described their engagement as noticing more details about specific elements of the environment: 'When we were assigned an important task, we

set up the war room that included the staff who were proficient in that work. Their comments would be considered as valuable information for planning and decisions.' (This represents mindful organizing—deference to expertise.) He discussed emphasizing the expert in the organization: 'We must evaluate the situation in the worst case ... Police are always ready for incidents; we are trained and always practice for unexpected situations.' (This represents organizational resilience—the management of keystone vulnerabilities.) He also spoke of participation in the police team's emergency management exercises.

'Our company was downsizing with employee layoffs and a bankruptcy process. We were forced to reduce costs. However, we used the remaining resources for creating a website, which we were the first in the printing media business, although, in that period, the internet was not ready and was difficult to use. Other media businesses did not create websites because they feared that revenue from selling magazines would be decreased.' (This represents individual mindfulness—novelty producing.) The executive editor of a media company described their solution when the organization went bankrupt from the impact of the economic crisis in 1997: 'Our company went bankrupt more than 10 years ago, and half of the staff were laid off. At the time, our income changed from paper to online. We do not think that it will recover. Now, staff returns to work with the same amount as lay off.' (This represents mindful organizing—a commitment to resilience.) He discussed mindful organizing and organizational resilience when faced with a crisis.

In Phase I, we found correlations between individual mindfulness, mindful organizing, and organizational resilience. To achieve all the research objectives, this study also employed quantitative methods in Phase II, including methodological triangulation or using more than one method to gather data, which increased the validity of the research results.

### 4.2. Quantitative Study

In this phase, we employed Structural Equation Modeling (SEM) to analyze a measurement model and structural model. It is a combination of confirmatory factor analysis and multiple regression. There are two parts of SEM in this study, including a measurement model and a structural model. The measurement model or confirmatory factor analysis (CFA) was used to test the reliability of the observed variables for those latent constructs in the hypothesized model. The structural model was based on estimating the relationship between the hypothesized latent constructs [36].

### 4.2.1. Sample and Procedures

The sample consisted of 685 individuals, including executives, managers, and employees from 10 organizations in Thailand. The respondents who experienced an organizational crisis were administered the survey developed for this study. The data collection period in this study was from June to December 2021. The researcher collected 670 questionnaires. After the data collection process, data cleaning was conducted to ensure that the data were correct, consistent, and usable. Thirty-one samples were removed, as questionnaires were incomplete, incorrect, or outlier cases. A total of 639 usable questionnaires were prepared for analysis in this study. The SPSS program was employed to check data according to the main assumptions of multiple regression analysis: linearity, normality, and homoscedasticity. A correlation matrix was checked to confirm that the Pearson correlations for all the variables made sense. The study participants were from three investigation divisions of the Royal Thai police (309), manufacturing (151), service (52), universities (41), hospitals (30), retailers (21), private schools (20), and gas terminal stations (15).

#### 4.2.2. Measures

The questionnaire consisted of three parts; each is described below.

*Individual mindfulness* was measured by the Langer Mindfulness Scale (LMS14). The original scale consisted of 21 items and four subscales: novelty producing, novelty seeking, flexibility, and engagement [37]. However, the researchers argued that certain scale items were not reliable and that the original four-factor model was indicative of a poor fit. Pirson and his colleagues developed a new model (i.e., LMS14), which consisted of 14 items and three subscales (i.e., by removing the flexibility subscale) [38,39]. The widely used scale is a reliable and valid measure of mindfulness that has been translated into many languages. All translated versions showed high internal consistency (i.e., Cronbach's alpha of 0.76 for the Persian version, 0.82 for the German, 0.83 for the Italian, and 0.78 for the Malaysian). The items were measured on a seven-point Likert scale ranging from 1 = disagree to 7 = strongly disagree, and certain reverse-coded items with higher scores indicated higher mindfulness.

*Mindful organizing* was measured using the Mindfulness Organizing Scale (MOS). Weick and Sutcliffe analyzed the mindful characteristics of HROs using the five principles to develop a 48-item questionnaire, which is a shorter form of the nine-item MOS questionnaire. They were found to have high internal reliability and to reflect theoretically derived and empirically observed content domains. Cronbach's alpha, which supported the reliability of the scale, was 0.88. Convergent validity was tested using a confirmatory factor analysis (CFA) and demonstrated an excellent fit across all fit indices (CFI = 0.964, incremental fit index = 0.964), and the discriminant validity of theoretically related constructs was approved. All nine survey items were measured using a five-point Likert scale from 1 = not at all to 5 = extremely in order to indicate the character of an organization.

*Organizational resilience* was measured by the BRT-13b, which is a short form of the benchmark resilience tool (BRT53). The BRT53 is a 53-item questionnaire developed by McManus (2008). A factor analysis was applied to the instruments, and CA for this scale was 0.95. However, using the BRT-53 for the survey revealed the scale's practical limitations, as the combined questionnaire was too long. Whitman et al. developed a short version of the BRT-53 to decrease survey time and improve response rates without significant losses in validity or reliability [40]. The BRT-13b was highly correlated to the BRT-53 overall resilience score, with all r-values exceeding 0.9 and showing significance. The reliability of the BRT-13b was assessed using CA at 0.85, which approved this scale. This measure consists of 13 items and two subscales (i.e., five for planning and eight for adaptive capacity), using a five-point Likert scale, from 1 = strongly disagree to 5 = strongly agree.

### 5. Results

#### 5.1. Factor Analysis

All the constructs were subjected to purification using exploratory and confirmatory factor analysis. First, SPSS was used to check whether the measures were above the minimum standard for conducting a factor analysis by the Kaiser–Meyer–Olkin (KMO) test and Bartlett's test of sphericity. Second, an exploratory factor analysis (EFA) of all scale items was considered, and high factor loadings of each instrument were selected. Cronbach's alpha, composite reliability (CR), and average variance extracted (AVE) were then examined. Subsequently, confirmatory factor analysis (CFA) was employed to assess the validity of the measures before entering the data to explore the correlation of constructs in the structural equation model.

*Individual mindfulness*. The factor analysis of the LMS14 extracted two components (i.e., novelty producing and seeking and engagement). The assessment of convergent validity was examined based on factor loadings, CR, and AVE. Four novelty-producing and seeking items (i.e., NP2, NP3, NS2, and NS3) and two engagement items (i.e., E1 and E3) were selected for high factor-loading indicators and consistency. The factor analysis of the six items showed that they explained 71.57%. According to Cronbach's alpha, the total reliability was 0.768. Using CFA showed a good fit, where $X^2$ (8, n = 639) = 38.824.

$X^2/df$ = 4.853, TLI = 0.956, CFI = 0.977, RMSEA = 0.078, and SRMR = 0.039. The AVE was used to assess the convergent validity, which at 0.586 was > 0.5. The CR was 0.892, which exceeded the recommended value of 0.7.

*Mindful organizing.* Four items (MO3, MO4, MO5, and MO9) from the MOS were selected, and they showed 66.82%. Cronbach's alpha was 0.833, which confirmed the total reliability. Using CFA showed a good fit, where $X^2$ (2, n = 639) = 3.933. $X^2/df$ = 1.966, TLI = 0.994, CFI = 0.998, RMSEA = 0.039, and SRMR = 0.007. The AVE was 0.561, and the CR was 0.835.

*Organizational resilience.* The factor analysis of the BRT-13b extracted two components (i.e., planning and adaptive capacity). Three planning items (i.e., OR1, OR3, and OR5) and two adaptive capacity items (i.e., OR10 and OR13) were selected for high factor-loading indicators and consistency. The factor analysis of the five items conveyed that they explained 75.40%. The total reliability, according to Cronbach's alpha, was 0.808. Using CFA showed a good fit, where $X^2$ (4, n = 639) = 10.788. $X^2/df$ = 2.697, TLI = 0.985, CFI = 0.994, RMSEA = 0.052, and SRMR = 0.008. The AVE was 0.597, and the CR was 0.880.

### 5.2. Structural Model and Hypotheses Testing Results

Structural equation modeling was employed to test the hypotheses for this study using AMOS 26 software with maximum-likelihood estimation. Overall, the combination of the independent variables explained 69.7% of the variance in organizational resilience. This study shows that the $X^2$ (83, n = 639) = 240.913. The ratio of relative chi-square (2.903) was valid. As Marsh and Hocevar suggested, this ratio should be 2.00–5.00 [41]. The RMSEA of the model in this study was 0.055; researchers have suggested that an RMSEA value less than 0.05 indicates a good fit and that values between 0.05–0.08 indicate a fair fit [42,43]. The SRMR was 0.036, and a value less than 0.08 presented a well-fitting model, whereas the NNFI (TLI) was 0.949, which is above 0.90 [43,44]. The CFI of this study was 0.959; Hu and Bentler suggested that a value of CFI $\geq$ 0.95 is indicative of a good fit [45]. In conclusion, all the measurements indicated that the model had a good fit (see Table 1).

**Table 1.** Model fit summary.

| Fit Indices | Model Value | Cut-Off Value | References |
|---|---|---|---|
| $X^2/df$ | 2.903 | 2.00–5.00 | Marsh & Hocevar (1985) [41] |
| RMSEA | 0.055 | <0.05 good fit<br>0.05–0.08 fair fit<br>0.08–0.10 mediocre fit<br>>0.10 poor fit | MacCullum et al. (1996) [42];<br>Hu & Bentler (1999) [43] |
| SRMR | 0.036 | <0.08 | Hu & Bentler (1999) [43] |
| NNFI (TLI) | 0.949 | >0.90 | Byrne (1994) [44];<br>Hu & Bentler (1999) [43] |
| CFI | 0.959 | $\geq$0.95 good fit | Hu & Bentler (1999) [43] |

Hypotheses 1 and 2 were supported (see Table 2). The statistics indicate that individual mindfulness has a positive effect on mindful organizing and that mindful organizing has a positive effect on organizational resilience. However, H3 was not supported, as individual mindfulness had no significant direct effect on organizational resilience. Figure 4 presents the structural model results.

From the significant direct effects found in H1 and H2, we further examined the mediating role of mindful organizing on the relationship between individual mindfulness and organizational resilience (H4). The mediating effect refers to a situation where a third variable intervenes between two associated constructs [45]. Before the SEM process, the conditions to claim the occurrence of mediation were tested [46]. In SEM, Hair et al. suggested the following steps for testing mediation [47]. We tested the first model by estimating the direct effect between individual mindfulness and organizational resilience. The result was positive and significant ($\beta$ = 0.44, $p < 0.01$). We then tested a second model

by adding mindful organizing to the first model. After adding mindful organizing to a model, the standard coefficient beta of the relationship between individual mindfulness and organizational resilience was not statistically significant and was close to zero ($\beta = 0.08$, $p = 0.284$). Therefore, potentially, mindful organizing fully mediates the path between individual mindfulness and organizational resilience, meaning H4 is supported.

**Table 2.** Hypotheses testing results.

| Hypotheses | β [a] | SE [b] | CR. [c] | p [d] | Result |
|---|---|---|---|---|---|
| H1 Individual mindfulness → Mindful organizing | 0.485 | 0.157 | 4.433 | 0.000 | Support |
| H2 Mindful organizing → Organizational resilience | 0.793 | 0.044 | 9.923 | 0.000 | Support |
| H3 Individual mindfulness → Organizational resilience | 0.080 | 0.065 | 1.071 | 0.284 | No |

[a] Standardized parameter, [b] standard error, [c] critical ratio, [d] significance level.

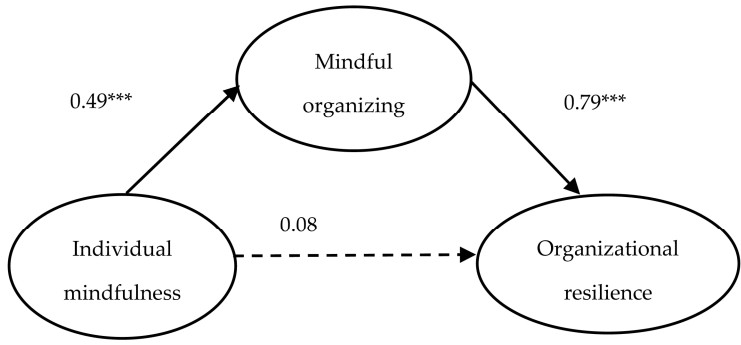

**\*\*\* Significant level at *p* ≤ 0.001**

**Figure 4.** Structural model result.

## 6. Discussion of the Findings

This study concentrated on the correlation between individual mindfulness, mindful organizing, and organizational resilience. Acting mindfully results in successfully managing unexpected events [20]. An exploratory mixed-method design was employed to investigate and measure the constructs in the model.

In Phase I, the qualitative study via the CIT confirmed the theoretical framework. The findings revealed that mindfulness—both leader mindfulness at the level of the individual (i.e., novelty producing, novelty seeking, and engagement) and mindful organizing at the level of the organization (i.e., preoccupation with failure, reluctance to simplify, sensitivity to operations, commitment to resilience, and deference to expertise)—acts as a key success factor of organizational resilience in times of crisis.

The findings in the quantitative study of Phase II confirmed that individual mindfulness has a positive impact on mindful organizing (H1). Previous studies have suggested a relationship between individual mindfulness and mindful organizing that focuses on five characteristics of HROs [23,30]. The concept of mindful organizing emerged from Langer's research on individual mindfulness; Langer suggested that mindfulness at the individual and organizational levels can be characterized correspondingly [15,16]. Weick and Robert suggested that when there are heedful relationships between individuals and when mindful comprehension is increased, organizational errors decrease [19]. The findings in this study presented that in a mindful state, individuals tend to be focused, maintain attention to a particular object, and remain more open to viewing the problem from a new perspective. When organizations face unexpected events, mindful individuals are better able to notice small failures, not simplify habitually, be sensitive to operations, focus on resilience, and shift locations of expertise.

Previous studies have presented the relationship between mindful organizing and organizational resilience. Shela et al. presented the vital role of collective mindfulness

or mindful organizing as capable of enhancing organizational resilience [48]. Wang et al. suggested that mindful organizing impacts organizational resilience processes in two megaprojects [31]. However, most studies emphasize qualitative methods, but there is limited empirical evidence to prove this relationship. Mindful organizing is capable of detecting and correcting failures and unexpected circumstances [9]. Commitment to resilience is one of the characteristics of mindful organizing in HROs, which develops the ability to recover from inevitable errors and maintain the functioning of the system. This study found that mindful organizing has a positive impact on organizational resilience (H2).

Although individual mindfulness has no significant direct effect on organizational resilience (H3), the current paper tested the mediating effect of mindful organizing on the relationship between individual mindfulness and organizational resilience. The findings suggested that mindful organizing partially mediates the effect of individual mindfulness on organizational resilience (H4).

It is concluded that individual mindfulness has no direct effect on organizational resilience, but individual mindfulness leads to increased mindful organizing, which in turn, leads to organizational resilience. Therefore, organizational resilience can be enhanced through mindful organizing.

## 7. Contributions of the Study

This study provides insight into how executives handle and recover from unexpected circumstances. Mindfulness as a predictive factor of organizational resilience will be useful for developing new knowledge of organizational resilience in the future. The theoretical, empirical, and practical contributions of the present study are as follows:

### 7.1. Theoretical Contributions

This study contributes to the theory of organizational resilience. The results indicate how organizations and individuals develop their capabilities to maintain function and structure in the face of adversity. This study is the first attempt to explore the role of mindfulness in enhancing organizational resilience through the core elements of individual mindfulness and mindful organizing. We found a direct effect of mindful organizing on organizational resilience. Moreover, the role of mindful organizing as a mediator has not previously been addressed. According to the findings of this study, mindful organizing mediates the relationship between individual mindfulness and organizational resilience.

### 7.2. Practical Contributions

The study has practical implications for research on managerial relevance. It contributes information regarding how individual mindfulness influences mindful organizing and enhances organizational resilience. The results of this study could enable entrepreneurs and management teams to understand the process of organizational resilience, improve organizational resilience through mindfulness, and, thus, enhance the sustainability of a business.

To be successful in enhancing mindfulness in an organization, the management team should (1) set mindfulness infrastructure as a core value and policy of the organization and communicate this value to all levels of employees as organizational culture; (2) design a mindfulness development system which links to work efficiency; (3) arrange an organizational environment, physical space, and work design which supports mindfulness development at the individual, team, and organizational levels; and (4) develop a mindfulness process in an organization which links individual mindfulness, team mindfulness, and mindful organizing. Such intervention may improve organizational resilience via mindfulness and other factors, which leads to the ability to prepare for, cope with, and recover from adverse situations and to continue to develop when handling adversity or crises. The management team who understands and applies this process in the organization can improve organizational resilience for the sustainability of the business.

## 8. Limitations and Future Research Directions

This research has limitations that must be noted. Firstly, this study examined mindfulness and organizational resilience in organizations exclusively in Thailand. The results may be limited if generalized to other countries. People of diverse cultures view the world through different cultural lenses. Thailand is an Eastern Buddhist country whose mindfulness perspective is different from that in modern Western contexts. Second, mindfulness may not be stable but rather changeable over time (i.e., changing through experience when faced with a crisis). This study employed a cross-sectional survey that examined the relationship between mindfulness and resilience in different population groups at a single point in time. However, it does not consider changes in mindfulness and resilience levels over a period of time.

In the present work, mindfulness and resilience research exhibits a growing body of qualitative and quantitative studies, but more work must be performed to expand its reach to organizational behavior. To learn more about organizational environments that enhance mindfulness and resilience, future research should consider organizational culture and structural or leader characteristics associated with higher levels of mindfulness and organizational resilience. Moreover, future research could explore appropriate training and activities to develop suitable mindfulness and resilience courses for diverse organizations in order to understand the impact of mindfulness on resilience better and to address how collective mindfulness shapes organizational resilience over time. Longitudinal studies extend beyond a single moment in time and maybe a fruitful area for further research. They are worth exploring to understand mindfulness and resilience to maximize their benefits for organizations in the future.

**Author Contributions:** Conceptualization, S.B. and W.L.; methodology, S.B. and W.L.; software, W.L.; validation, S.B., W.L. and N.J.; formal analysis, W.L.; investigation, W.L.; resources, W.L.; data curation, W.L.; writing—original draft preparation, W.L.; writing—review and editing, S.B., N.J. and K.C.; visualization, W.L.; supervision, S.B., N.J. and K.C. All authors have read and agreed to the published version of the manuscript.

**Funding:** This research received no external funding.

**Institutional Review Board Statement:** The study was conducted in accordance with the Declaration of Helsinki and approved by the Ethics Committee of CHIANG MAI UNIVERSITY (protocol code CMUREC 63/197, date of approval: 6 October 2022).

**Informed Consent Statement:** Informed consent was obtained from all subjects involved in the study.

**Data Availability Statement:** Data available on request due to privacy restriction.

**Acknowledgments:** We thank all the participants who attended the interviews and helped with this research.

**Conflicts of Interest:** The authors declare no conflict of interest.

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
