# Peer review of "Enhancing Organizational Resilience through Mindful Organizing"

_sustainability, doi:10.3390/su15032681_

Round 1
Reviewer 1 Report
The paper adopts a combination of qualitative and quantitative research methods to explore insights into mindfulness and resilience at the organizational level. The demonstration process is relatively complete. However, the discussion and conclusion are too simple, which requires in-depth analysis and discussion of the empirical results.
Author Response
Response to Reviewer 1 Comments
We appreciate you for your precious time in reviewing our paper and providing valuable comments. It was your valuable and insightful comments that led to possible improvements in the current version. We have carefully considered the comments and tried our best to address all your comments. We have green highlighted the changes within the manuscript attached.
Point 1: The paper adopts a combination of qualitative and quantitative research methods to explore insights into mindfulness and resilience at organizational level. The demonstration process is relatively complete. However, the discussion and conclusion are too simple, which requires in depth analysis and discussion of the empirical results.
Response 1: We added the discussion and conclusion in the section of Discussion of the Findings (p.11-12) as follows;
This study concentrated on the correlation between individual mindfulness, mindful organizing, and organizational resilience. Acting mindfully results in successfully managing unexpected events [20]. An exploratory mixed-method design was employed to investigate and measure the constructs in the model.
In Phase I, qualitative study via the CIT confirmed the theoretical framework. The findings revealed that mindfulness—both leader mindfulness at the level of the individual (i.e., novelty producing, novelty seeking, and engagement) and mindful organizing at the level of the organization (i.e., preoccupation with failure, reluctance to simplify, sensitivity to operations, commitment to resilience, and deference to expertise)—acts as a key success factor of organizational resilience in times of crisis.
The findings in the quantitative study of Phase II confirmed that individual mindfulness has a positive impact on mindful organizing (H1). Previous studies have suggested a relationship between individual mindfulness and mindful organizing that focuses on five characteristics of HROs [23, 30]. The concept of mindful organizing emerged from Langer’s research on individual mindfulness; Langer suggested that mindfulness at the individual and organizational levels can be characterized correspondingly [15, 16]. Weick and Robert suggested that when there are heedful relationships between individuals and when mindful comprehension is increased, organizational errors decrease [19]. The findings in this study presented that in a mindful state, individuals tend to be focused, maintain attention on a particular object, and remain more open to viewing the problem from a new perspective. When organizations face unexpected events, mindful individuals are better able to notice small failures, not simplify habitually, be sensitive to operations, focus on resilience, and shift locations of expertise.
Previous studies have presented the relationship between mindful organizing and organizational resilience. Shela et al. presented the vital role of collective mindfulness or mindful organizing as capable of enhancing organizational resilience [48]. Wang et al. suggested that mindful organizing impacts on organizational resilience processes in two megaprojects [31]. However, most studies emphasize qualitative methods, but there is limited empirical evidence to prove this relationship. Mindful organizing is capable of detecting and correcting failures and unexpected circumstances [9]. Commitment to resilience is one of the characteristics of mindful organizing in HROs, which develop the ability to recover from inevitable errors and maintain the functioning of the system. This study found that mindful organizing has a positive impact on organizational resilience (H2).
Although individual mindfulness has no significant direct effect on organizational resilience (H3), the current paper tested the mediating effect of mindful organizing on the relationship between individual mindfulness and organizational resilience. The findings suggested that mindful organizing partially mediates the effect of individual mindfulness on organizational resilience (H4).
It is concluded that individual mindfulness has no direct effect on organizational resilience, but individual mindfulness leads to increased mindful organizing, which in turn, leads to organizational resilience. Therefore, organizational resilience can be enhanced through mindful organizing.

Reviewer 2 Report
Dear authors,
Thank you very much for sending me your article to review. You have done research on alluring topics. Please find the enclosed file.
Kind regards,
Reviewer.

Author Response
Response to Reviewer 2 Comments
We appreciate you for your precious time in reviewing our paper and providing valuable comments. It was your valuable and insightful comments that led to possible improvements in the current version. We have carefully considered the comments and tried our best to address all your comments. We have blue highlighted the changes within the manuscript attached.
Results section.
Point 1: The result section should stand independent or together with the discussion section, but not with the methodology section. The authors can make the results section separated from the methodology section.
Response 1: The results section has already separated from methodology section and stand independent in section 5 (p. 9).
Point 2: The survey period in this cross-sectional study is loose. The authors can explain this in section 4.3.1 (Sample and procedures).
Response 2: The survey period has already added in section 4.3.1 (p. 8). “The data collection period in this study was from June to December 2021.”
Point 3: Figure 4 is an output from AMOS for the SEM analysis. The authors should be able to draw proper models within AMOS. There are 3 main unobserved variables in this study, not five.
Response 3: Firstly, we would like to present an output from AMOS presented that there are multiple regression of 3 main constructs; individual mindfulness, collective mindfulness, and organizational resilience, and confirmatory factor analysis of individual mindfulness (two components of novelty seeking and producing, and engagement), and organizational resilience (two components of planning, and adaptive capacity).
However, we modified Figure 4 Structural model result that are three main unobserved variables in this study as suggested (p.10).
Point 4: In Figure 4, the points (.XX) before the two results numbers is not appropriate for presentation. Authors may present a null (0) before the period. For example, 0.65 instead of .65.
Response 4: We have already modified and presented the structural model result in Figure 4. with a null before the period as suggested (p.10).
Literature Review section.
Point 5: The authors can provide the source for Figure 1.
Response 5: The source for Figure 1 was added “Adapted from Weick and Sutcliffe, 2001 [21]” (p. 3).
Point 6: Figure 2 is not consistent with hypothesis 3. The authors can provide clarification on this.
Response 6: We added the line that link between individual mindfulness and organizational resilience in Figure 2 (p. 4) as suggested.
Point 7: On the one hand the authors build hypotheses based on your findings in phase 1 with the deductive method. On the other hand, section 4.2 (Research Model and Hypotheses) should be in the literature review section. Can the authors come up with an explanation for this uncommon confusion?
Response 7: We have already moved section 4.2 Research model and Hypotheses to section 3.2 Research model and Hypotheses (p.5).
Point 8: The paragraphs before the hypotheses are explained, do not elaborate the thinking about the relationship that becomes the hypotheses. So, the hypotheses are not robust. The authors can use relevant references in explaining these relationships.
Response 8: Before the hypotheses are explained, we referred relevant works in explaining relationships and added relevant research as you suggested (p.5) as;
Hypothesis 1. Mindful organizing was developed in individual mindfulness as a foundation. It is a social process which becomes a collective capability through interactions among individuals [15, 22]. Individual and mindful organizing share an emphasis on increased attention to the present moment situation and acting on what they notice [9, 23, 30]. This study attempts to theoretically confirm the link between individual mindfulness and mindful organizing.
Hypothesis 2. Mindful organizing is the quality of attention at the level of the collective organization which relates to what people decide to do with what they notice when facing unexpected events [9]. At the organizational level, five characteristics of HROs were identified as the necessary elements in organizational resilience [31, 32]. Oliver et al. define organizational mindfulness as a quality of an organization which reliably and effectively operates in the face of challenging conditions, and they find a significant positive correlation between mindfulness, resilience, and performance [33]. This study explored the relationship between mindful organizing and organizational resilience to contribute to the theory of HROs.
Hypothesis 3. There is no empirical evidence of the relationship between individual mindful-ness and organizational resilience. However, certain arguments exist regarding the positive impact of mindfulness on organizational change, sustainability, outcomes and performance, leaders’ decision making, and success [7, 22, 23, 24].
Hypothesis 4. Most studies on mindfulness at the collective organizational level have been qualitative [11]. Hence, no evidence indicates mindful organizing as a mediator of con-structs. There is a need for more research on mindful organizing at work, especially using quantitative methods. This study is the first attempt to explore the relationship between individual mindfulness and organizational resilience by using mindful organizing as a mediator.
Introduction section.
Point 9: The first sentence in the fourth paragraph “However, fewer studies have…………resilience.” Please provide examples as well as references.
Response 9: We provided reference [10] in sentence “However, fewer studies have emphasized the association between mindfulness and organizational resilience [10].” (p.2).
Work of “Liu X, Wu X, Wang Q, Zhou Z. Entrepreneurial mindfulness and organizational resilience of Chinese SMEs during the COVID-19 pandemic: The role of entrepreneurial resilience. Frontier Psychology. 2022.”
Point 10: The third sentence in the fourth paragraph “Therefore, this study’s goal of is ……………organizing.” Where is the explanation about Mindful Organizing?
Response 10: The explanation about mindful organizing is at last sentence at the first paragraph (p.2). as;
“Mindful organizing is the collective capacity of members in an organization to attend to and act on errors and unexpected circumstances [9].”
Point 11: Sentences 1 to 4 are not meaningful to include.
Response 11: We have eliminated sentences 1 to 4 in abstract as suggested.
Title.
Point 12: The title does not represent the issues raised in the research (research questions). For this reason, the authors should make a little that is consistent with the hypotheses and research framework. “Individual mindfulness and Organizational resilience: The mindful organizing moderation”
Response 12: We would like to request you to remain the title as “Enhancing organizational resilience through mindful organizing”
To present that this title represents the issues raised in the research, we added these sentences as;
1.In section of Introduction (p.2, paragraph 2, sentence 3) specify the objective of this study and research question as;
“Therefore, this study aims to explore the correlation between mindfulness and organizational resilience, and whether the impact of mindfulness at the individual and/or organizational level can enhance organizational resilience.”
2. In section of discussion of findings (P.12, last sentence), we answer research question as;
“It is concluded that individual mindfulness has no direct effect on organizational resilience, but individual mindfulness leads to increased mindful organizing, which in turn, leads to organizational resilience. Therefore, organizational resilience can be enhanced through mindful organizing”

Reviewer 3 Report
bibliographical references should be improved. put more quotes from the last 5 years.section 4.3 needs to be rewritten, citing in detail the modeling used; explaining the mathematical models
I suggest this papers:
Santos, Davidson de Almeida ; Quelhas, Osvaldo Luiz Gonçalves ; Gomes, Carlos Francisco Simões ; FILHO, JOSÉ RODRIGUES DE FARIAS . Theoretical Proposal for an Integrated Sustainability Performance Measurement System in the Supply Chain. Frontiers in Sustainability, v. 2, p. 720-763, 2021.
Author Response
Response to Reviewer 3 Comments
We appreciate you for your precious time in reviewing our paper and providing valuable comments. It was your valuable and insightful comments that led to possible improvements in the current version. We have carefully considered the comments and tried our best to address all your comments. We have yellow highlighted the changes within the manuscript attached.
Point 1: Bibliographical references should be improved, put more quotes from the last 5 years.
Response 1: We added relative references from the last 5 years in section of References (p.13-15) as follows;
Liu X.; Wu X.; Wang Q.; Zhou Z. Entrepreneurial mindfulness and organizational resilience of Chinese SMEs during the COVID-19 pandemic: The role of entrepreneurial resilience. Front Psychol. 2022 Oct 6; 13:992161. http://org/doi: 10.3389/fpsyg.2022.992161.
Wang, L.; Müller, R.; Zhu, F.; Yang, X. . Collective Mindfulness: The Key to Organizational Resilience in Megaprojects. Proj. Manag. J. 2021, 52, 592–606. https://doi.org/10.1177/87569728211044908
Shela, V.; Ramayah, T.; Ahmad, N.H. inducing organizational resilience through collective mindfulness: A path towards an uninterrupted metamorphosis. Dev. Learn. Organ. 2022, 36, 4-6. https://doi.org/10.1108/DLO-08-2021-0147
Point 2: Section 4.3 needs to be written, citing in detail the modeling used; explaining the mathematical models
Response 2: We added these sentences to (1) citing in detail the structural equation modeling (p.8), and (2) explain the mathematical model (p.5) as;
(1) citing in detail the structural equation modeling;
In this phase, we employed Structural Equation Modeling (SEM) to analyze a measurement model and structural model. It is a combination of confirmatory factor analysis and multiple regression. There are two parts of SEM in this study, including a measurement model and a structural model. The measurement model or confirmatory factor analysis (CFA) were used to test the reliability of the observed variables for those latent constructs in the hypothesized model. The structural model was based on estimating the relationship between the hypothesized latent constructs [36].
(2) Explain the mathematical model;
Figure 3 shows a path diagram for the causal relationships between the three con-structs in enhancing organizational resilience: individual mindfulness (ξ1), mindful organizing (η1), and organizational resilience (η2). Mindful organizing and organizational resilience are endogenous latent variables, while individual mindfulness is an exogenous latent variable. The SEM for this mediation model is given by:
η1 = β ξ1 η1 ξ1 + ζ η1
η2 = β η1 η2 η1 + β ξ1 η2 ξ1 + ζ η2
Point 3: I suggest this paper: Santos, Davison de Almeida; Quelhas, Osvaldo Luiz Goncalves; Gomes, Carlos Francisco Simoes; Filho Jose Rodrigues De Farias. Theoretical proposal for an Integrated Sustainability Performance Measurement System in the Supply Chain. Frontiers in Sustainability, v.2, p.720-763, 2021
Response 3: Thank you for your suggestion. This paper is very interesting for sustainability in the supply chain. However, it is not relevant for our research. In the future, we are pleased to cite this paper in our research if possible.

Round 2
Reviewer 2 Report
Dear authors,
Thank you for sending me a revised version of your article. Now I see your hard work. Before it is accepted, I advice you to improve the English style for the article. Also, Figure 2 is not important to be presented as a repetition.
Many thanks.
Author Response
Dear Sir;
We would like to take this opportunity to thank you again for the effort and expertise that you contributed toward reviewing our manuscript.
We have already improved our manuscript as you suggested as;
Point 1. Improve the English style of article.
Response 1. We have already sent our manuscript to the MDPI Editing service for English language as the certificate attached.
Point 2. Figure 2 is not important to be presented as a repetition.
Response 2. We would like to explain that we intend to show Figure 2 for illustrating the core elements of three constructs in theoretical framework to make it more clear and understand easily for reader. In Figure 1, we presented only the construct of mindful organizing and meaning of five of characteristics of mindful organizing. In Figure 3, we presented only the conceptual model and hypotheses. In Figure 4, we presented the results of model. Therefore, we would like to confirm to remain Figure 2 in this manuscript. Would you please consider about this.
Many thanks again for your kind support. We very much appreciate your help.
Yours sincerely,
Wiphawan Limphaibool

Reviewer 3 Report
I suggest including this paper:
Santos, Davidson de Almeida ; Quelhas, Osvaldo Luiz Gonçalves ; Gomes, Carlos Francisco Simões ; FILHO, JOSÉ RODRIGUES DE FARIAS . Theoretical Proposal for an Integrated Sustainability Performance Measurement System in the Supply Chain. Frontiers in Sustainability, v. 2, p. 720-763, 2021.
Author Response
Dear sir;
Thank you so much again for your thoughtful comments and efforts towards
improving our manuscript.
We try to read the paper "Theoretical Proposal for an Integrated Sustainability Performance Measurement System in the Supply Chain." that you suggested many times for link this paper with our manuscript. Although, this paper is valuable and interesting, however, we have not found any point that relate to our manuscript. Therefore, we can not include this paper in our reference. We would like to apologize for the inconvenience. Please consider our reason.
Thank you so much again for you kind support.
Yours sincerely,
Wiphawan Limphaibool